# Obesity-Associated Anxiety Is Prevalent among College Students and Alleviated by Calorie Restriction

**DOI:** 10.3390/nu14173518

**Published:** 2022-08-26

**Authors:** Junbo Wang, Xinyi Ran, Jinchen Ye, Run Deng, Weimin Dang, Yangyi Fan, Zhiping Hu, Lei Yang, Wentian Dong, Yifei Lv, Kunzhan Lin, Maoran Li, Yuhe Jiang, Ruimao Zheng

**Affiliations:** 1Department of Anatomy, Histology and Embryology, School of Basic Medical Sciences, Health Science Center, Peking University, Beijing 100191, China; 2Department of Basic Medical Sciences, School of Medicine, Tsinghua University, Beijing 100084, China; 3Peking University Institute of Mental Health, Peking University, Beijing 100191, China; 4NHC Key Laboratory of Mental Health, Peking University, Beijing 100191, China; 5National Clinical Research Center for Mental Disorders, Peking University Sixth Hospital, Beijing 100191, China; 6The Division of Medical Affairs, Peking University Sixth Hospital, Beijing 100191, China; 7Department of Neurology, Peking University People’s Hospital, Beijing 100044, China; 8Department of Pathology, School of Medicine, University of Pittsburgh, Pittsburgh, PA 15260, USA; 9The Division of Psychosomatic Medicine, Peking University Sixth Hospital, Beijing 100191, China; 10Department of Prosthodontics, Peking University School and Hospital of Stomatology, Beijing 100081, China; 11Neuroscience Research Institute, Peking University, Beijing 100191, China; 12Key Laboratory for Neuroscience of Ministry of Education, Beijing 100191, China; 13Key Laboratory for Neuroscience of National Health Commission, Beijing 100191, China

**Keywords:** obesity, anxiety, college students, calorie restriction, sleep duration, dieting

## Abstract

Anxiety is a common disorder among college students, especially those with obesity. Obesity contributes to metabolic disorders and disturbs the neural functions, further leading to anxiety. In this cross-sectional study, we aimed to determine the association between obesity and anxiety among college students and identified the potential factors for obesity-associated anxiety. We evaluated the intervention effects of calorie restriction on anxiety. Self-reported questionnaires were distributed to 1381 college students from January to March in 2021. Anxiety was measured by the State-Trait Anxiety Inventory (STAI). Participants were classified into anxiety and non-anxiety groups according to their STAI scores. Chi-squared test and logistic regression were used to analyze the potential factors. We found that 383 college students exhibited anxiety, accounting for 30.1% among all included college students, which was higher than the global average. The association between anxiety and obesity was observed among college students (*p* = 0.009), especially in males (*p* = 0.007). We identified that pre-obesity (*p* = 0.012), unhealthy calorie intake (*p* = 0.001), dieting (*p* = 0.003) and high academic year (*p* = 0.006) as the risk factors for anxiety and found that the long sleep duration was a protective factor for anxiety (*p* < 0.001). We found that more obese students showed an improvement of anxiety than the underweight students after calorie restriction (*p* < 0.001). Collectively, our findings suggest that obesity-associated anxiety is prevalent among the college students and could be alleviated by moderate calorie restriction. It is necessary for students to receive anxiety management in their college life. Additionally, the proper calorie restriction should be promoted to help students protect against obesity and obesity-associated anxiety.

## 1. Introduction

As a common mental health disorder, anxiety disorders are typically characterized by hyperarousal, excessive fear and worry [1]. Patients with anxiety exhibit generalized anxiety disorder (GAD), social phobia, panic disorder, specific phobias, agoraphobia, separation anxiety disorder and selective mutism. In recent years, the prevalence of anxiety disorders is increasing worldwide, ranging from 3.8 to 25% across countries, which is even reaching 70% in people with chronic health conditions [2]. Results from the Global Burden of Disease study suggest that anxiety disorders are the sixth leading cause of global disability [3], accounting for 26.8 million disability-adjusted life years [4]. In addition, anxiety disorders are highly associated with elevated cardiovascular risk factors such as high blood pressure [5], obesity [6,7] and premature mortality [8,9].

Emerging evidence shows that the incidence of obesity in the US is 42% among adults, with severe obesity at 9%. In China, obesity has also reached 16.4% in adults [10] which is higher than the global average incidence [11]. Therefore, it is important for college students to prevent obesity as well as its associated disorders in adolescents. It is well recognized that obesity serves as a major risk factor for several diseases, such as anxiety [12] and depression [13]. For example, Gariepy et al. reported a positive association between anxiety and obesity through a systematic review and meta-analysis study [12]. Physiologically, the hypercaloric diets impair fear-regulatory mechanism which is evoked by alterations of synaptic plasticity in the prefrontal cortex, amygdala, and hippocampus; these regions underlie the formation of an essential neural circuits to regulate fear [14,15]. It has been reported that the rats that underwent hypercaloric food-intake from adolescence showed a protracted elevation of the corticosterone level responding to adverse stimulus; these rats also showed high levels of neuronal activation and synaptic plasticity within the basolateral nucleus of amygdala [16]; these observations uncovered that the hypercaloric food-intake in adolescence may be positively linked to development of anxiety disorder in adulthood [17]. Another report also shows that the increased rates of obesity promote the development of anxiety disorder in young people [18]. Taken together, these findings reflect that calorie restriction may be beneficial for the relief of anxiety. Nevertheless, the association between anxiety and obesity among college students remains elusive.

In this study, we aimed to identify the factors influencing obesity-associated anxiety among college students. Firstly, we accessed the association between anxiety and obesity of college students in China. Further, we explored the potential factors affecting anxiety, such as sleep duration, physical activity and dietary behaviors. Importantly, we evaluated the intervention effects of calorie restriction on anxiety. This study will help administrative departments to provide support for college students to improve their psychological health.

## 2. Materials and Methods

### 2.1. Ethics Statement

All participants were informed about the objectives of the study, and the questionnaire information items were explained. Each participant was assured by assigned with a code number for analysis. The study was approved by the Institutional Review Board of Peking University Sixth Hospital (No. 2021-11). All procedures were tightly followed the guidelines of the Declaration of Helsinki.

### 2.2. Participants

This is a cross-sectional study conducted from January to March in 2021 in Beijing, China. Students with diabetes, bulimia, anorexia, thyroid dysfunctions, drug/substance abuse, smoking and alcoholism were not included in this study. Over 1381 university students from Peking University were selected by utilizing stratified random sampling. We excluded 16 (0.1%) students that did not complete the questionnaire and 13 (0.1%) students with abnormal body mass index (BMI) (BMI < 10 or BMI > 50). In the questionnaire, we set up a general-knowledge question that “You are a human” to assess the reliability of the respondents. Nonetheless, 80 (5.8%) students selected “no”. After excluding these students with unbelievable information, 1272 (92.1%) were included in the final analysis to assess the association between anxiety and obesity and identify the factors influencing obesity-associated anxiety (Figure 1).

### 2.3. Measurements

The baseline variables included sex, BMI, academic year, major, social behaviors and questionnaires on anxiety self-test scales. Beck Anxiety Inventory (BAI) was used to assess the anxiety level among college students. The BAI contains a 21-item scale measuring the severity of self-reported anxiety. Anxiety symptoms were rated on a 4-point scale as follows: 0 = none; 1 = mildly; 2 = moderately; and 3 = severely [19]. Higher total scores indicate greater anxiety. 

On social behaviors, we detected the dietary habits, sleep duration and physical activity among these participants. The questionnaire on dietary habits includes calorie intake and food preference. We assessed the daily intake of beverages, vegetables, fruits, staples and meat of all students. According to their daily intake, participants were asked to assign a value of each food from 0–10, from which food preference could be also determined. High values imply more intake of food, which also indicates a food preference. Sleep duration was determined by asking the question “How long do you sleep on weekdays?”; the possible answers were “<4 h”, “4–6 h”, “6–8 h” and “≥8 h”. 

Physical activity was assessed by a physical activity questionnaire which evaluates levels of physical activities during exercise. The questionnaire consists of 3 items and records physical activity at four intensity levels. Firstly, the exercise frequency was collected by asking “How often do you exercise in one week?”. Moreover, the participants were asked to report physical activities performed for at least 10 min during the last 7 days. The possible answers included “vigorous intensity activities such as running, swimming and rope skipping”, “moderate-intensity activities such as leisure cycling, yoga and dancing”, “slight-intensity activities such as walking and climbing stairs, and “no-intensity activities such as sitting”. Finally, the amount of time spent in physical activities was collected, which was calculated by considering the estimated metabolic equivalent (MET) for that activity. Dieting was defined as intentional restriction of calorie intake, which was determined by asking the question “Have you been on a dieting during the last two weeks?”, and the possible answers were “Yes” or “No”. 

According to the guidance of the World Health Organization (WHO), BMI was divided into four groups: underweight (BMI < 18.5), normal weight (18.5 ≤ BMI < 24.0), pre-obesity (24.0 ≤ BMI < 28.0) and obesity (28.0 ≤ BMI). Similarly, a total score of BAI was calculated to assess the anxious level. The total score of BAI that is higher than 45 indicates a severe anxiety of the respondents. Therefore, all students were classified into non-anxiety (total score < 45) and anxiety (45 ≤ total score). 

On dietary behaviors, the calorie intake was calculated after giving a weighting. According to its calorie, beverages, vegetables, fruits, staples and meat were weighted by 60, 20, 40, 100 and 140, separately. Then, the total score was calculated as the calorie intake of each student (kcal/d). Calorie intake was divided into two groups: healthy calorie intake (1600–2400 kcal/d) and unhealthy calorie intake (0–1600 kcal/d or > 2400 kcal/d). Similarly, physical activity was also weighted with the intensity of exercise. With the increased intensity of exercise, the weight score was 3.3, 4.0, 5.5 and 8.0. The physical activity was classified into three groups: 0–600 MET∙min/w, 600–3000 MET∙min/w and >3000 MET∙min/w. The major in medical science includes Basic Medical Sciences, Clinical Medical Sciences, Nursing, Pharmaceutical Sciences, Public Health or Dentistry, and all of the other majors were classified as non-medical science.

### 2.4. Statistical Analysis

Data were analyzed by using the Statistical Package for the Social Sciences (SPSS) 26.0. Descriptive analyses were conducted to determine the demographics, dietary habits and anxiety of college students. To determine the association between anxiety and obesity, chi-squared tests and univariate logistic regression were run among male and female students. To identify the factors affecting anxiety, the baseline characteristics were compared between students with anxiety and without anxiety using the chi-squared test and logistic regression. To assess the effect of calorie restriction, chi-squared tests were run on sex, obesity, academic year and calorie restriction between students with improved anxiety and non-improved anxiety. Statistical significance for two-tailed *p* values was defined as *p* < 0.05.

## 3. Results

The demographic characteristics of 1272 respondents were shown in Table 1. Males represented 42.9% of the participants and 57.1% were female. Approximately 16.6% of the respondents were underweight, 68.1% were normal weight, 12.2% were pre-obesity and 3.1% were obesity, according to previous study [20]. About 30.3%, 41.7%, 18.9%, 9.0% of the respondents responded that they were freshmen, sophomores, junior and senior. Most of the respondents were non-medical students (59.9%). One-third of the respondents (30.1%) were anxious. Approximately 14.5% of the respondents had enough sleep duration (>8 h/d). Only 13.2% of respondents stuck to dieting.

Emerging evidence revealed that there was a positive association between obesity and anxiety in adults. Here, in this study, we assessed this association among college students. Notably, we assessed the internal consistency of the STAI, and found that the Cronbach’s alpha was 0.94, which indicated a high reliability of the responses. As expected, the underweight students showed less severe anxiety than students with normal weight, pre-obesity and obesity, especially in males (*p* < 0.05) (Figure 2). Similarly, anxiety was positively associated with obesity among college students by using logistic regression, especially in males (Table 2), showing that the obese students were more likely to exhibit anxiety than the underweight students (odds ratio [OR]: 5.03, 95% confidence interval [CI]: 1.08–23.40).

Furthermore, we explored the potential factors influencing obesity-associated anxiety. A total of 8 potential variables were chosen as independent variables and analyzed by logistic regression separately. The results were presented in Table 3. Among them, 4 variables were significantly correlated to anxiety (*p* < 0.05). Compared with the non-anxious students, anxious students were likely to have higher BMI, unhealthy calorie intake, shorter sleep duration and be on the diet (*p* < 0.05) (Table 3). Specifically, the prevalence of anxiety was observed in 14 (35.0%) obese and 49 (23.2%) underweight students, respectively (*p* = 0.009) (Table 3). Anxious students were likely to have calorie intake more than 2400 kcal/d or less than 1600 kcal/d (*p* < 0.001). Moreover, the short sleep duration was positively associated with anxiety (*p* < 0.001). Students who were on the diet were more likely to be anxious (*p* = 0.002).

Unadjusted logistic regression models showed that pre-obesity (OR (95% CI): 2.02 (1.23–3.31)), unhealthy calorie intake (OR (95% CI): 1.57 (1.23–2.01)), dieting (OR (95% CI): 1.70 (1.22–2.38)), long sleep duration (OR (95% CI): 0.24 (0.11–0.53)) and junior academic year (OR (95% CI): 1.55 (1.09–2.19)) were significantly associated with anxiety, while anxiety was not associated with sex, major and physical activity (*p* > 0.05) (Table 4). Even after multivariable adjustment for relevant factors, pre-obesity (1.91 (1.16–3.17)), unhealthy calorie intake (1.56 (1.21–2.00)), dieting (1.68 (1.19–2.36)) and junior academic year (1.35 (1.00–1.83)) were at high risk for anxiety, while long sleep duration (0.21 (0.10–0.48)) was identified as a protective factor for anxiety (Table 5). Notably, in multivariate logistic regression model, the students with normal weight showed no observed high risk compared to these with underweight (*p* = 0.065) while this difference was significant in univariate logistic regression analysis (*p* = 0.036). Conversely, sophomore academic year was at a higher risk of anxiety compared to the freshman (*p* = 0.049) in multivariate logistic regression model while no significant difference was detected in univariate logistic regression analysis (*p* = 0.077) (Table 4 and Table 5).

In aforementioned evidence, we have identified obesity and unhealthy calorie intake as risk factors for anxiety. Therefore, we explored the effects of calorie restriction on the improvement of anxiety. A total of 230 subjects were randomly selected from all participants and received calorie restriction (3–7d), including 122 females and 108 males. Students who received calorie restriction were prescribed a low-calorie diet (400 kcal lower than caloric requirement per day). After the intervention, STAI scores were obtained at the same time of day (9:00–10:00 am). Then, the efficacy was evaluated by the difference value of STAI score before and after intervention. “>5” was regarded as an improvement of anxiety according to a reported study [21]. The results were shown in Table 6. Compared to underweight students, more students exhibited improved anxiety (*p* = 0.001). Moreover, the effects were comparable in students classified by academic year, dieting or calorie intake. Results showed that there were no differences of improved anxiety in senior academic year, dieting and unhealthy calorie intake compared to freshman (*p* = 0.727), no-dieting (*p* = 0.408) and healthy calorie intake (*p* = 0.418), respectively.

## 4. Discussion

In this study, we found an association between anxiety and obesity among college students. Obesity, unhealthy calorie intake, dieting and high academic year were identified as risk factors for anxiety while long sleep duration acted as a protective factor. In addition, our observation also demonstrated that after calorie restriction, the obese students were more likely to exhibit improved anxiety than the underweight students. These findings may generate greater momentum to encourage public health efforts to manage anxiety for college students. 

The prevalence of anxiety among college students was 30.1% in this study, which was appropriately 10 times more than the global average [22]. The difference in prevalence mirrored a higher prevalence of anxiety among college students, although similar questionnaires from different studies show different prevalence [23,24]. The high prevalence of anxiety may be attributed to drastic changes in the living environment. College students not only have heavy academic pressure, but also have to learn how to deal with daily life independently. In addition, the intense work pace and the complex interpersonal relationships may overwhelm the college students. As a result, the high incidence of anxiety disorders may prevail among the college students, which should be paid special attention to.

A previous study demonstrates that the obesity is positively related to promoting the development of anxiety [25]. In this study, the prevalence of anxiety in pre-obese students was 39.5%, which was higher than that in underweight students (23.2%). The results of logistic regression analyses also showed that the risk of anxiety was increased with the development of obesity, indicating a remarkable association between obesity and anxiety among college students. Interestingly, the obese students did not show a marked high risk of anxiety compared with the underweight students; this may be due to a few numbers of the obese students who were included in this study. In addition, anxiety profoundly affects the dietary behavior of college students. The high-anxiety group exhibited fewer healthy calorie intake as the students in this group tended to eat more sugar-based snacks, carbohydrate-rich food and fried food, as compared with the students in the low-anxiety group. This observation was in agreement with the findings of a previous research [26]. Moreover, students with unhealthy calorie intake or dieting were more likely to exhibit anxiety. It is reported that skipping meals is associated with anxiety and depressed mood among adolescents [27], suggesting that skipping meals could be both the cause and result of anxiety. Moreover, in patients with anorexia nervosa, pre-meal anxiety was significantly associated with reduced calorie intake and reduced consumption of calories from fat at the meal [28], demonstrating that severe anxiety was closely related with little calorie intake; this can explain why individuals who were on a diet showed more anxiety than those who were not on a diet. In addition, anxious adolescents are more likely to engage in binge eating, especially in females [29]. Strict limitations of foods may trigger overeating in the anxious individuals. Thus, both female and male students should be informed about healthy food consumption. Notably, we found that short sleep duration also acted as a risk factor for anxiety. It is reported that adolescents with sleeping less than 8.5 h/day in boys and less than 7.5 h/day in girls had elevated scores for anxiety symptoms [30]. The results suggest that healthy calorie intake and enough sleep duration may act as protective factors for college students against anxiety.

Differences in the students’ anxiety based on academic year suggest the stress of college students. A survey has shown that senior students display a high stress compared to freshman [31], which may indicate high anxiety of these senior students. In our study, we also found that more junior students showed greater anxiety compared with freshman, which might be attributed to the higher level of academic pressure. Additionally, it has been reported that the major of students may affect their anxiety levels. Honestly, we noticed that several studies showing that the non-medical students exhibit a significantly higher anxiety levels than the medical students [32,33]. It may be more professional medical knowledge and better health awareness in the medical students compared to the non-medical students, which is beneficial for prevention of anxiety. Nevertheless, in our research, the prevalence of anxiety among the non-medical students was 31.5% while the prevalence of anxiety was 28.0% among the medical students; and no statistical significance was observed. We speculated that this discrepancy may be due to the strengthening of health education for the college students in China recently. Hence, college administrations should offer assistance that supplies psychological guidance and professional medical knowledge for college students via leaflets or posters and provides psychological treatments for the students with high anxiety.

Calorie intake represents one of the major determinants of mental health and the vulnerability to injury and disease. Evidence has showed that long-term high fat diet increases serum corticosterone levels and hippocampal expression of CYP11B1, which elevates HPA activity and then induces anxiety [34]. In addition, the gut microbiota influences obesity-related gut- and intestinal-based metabolic factors and affects anxiety-like behavior via the microbiota-gut-brain axis [35,36], which could partly explain the co-morbidity of obesity and anxiety-like behavior. Notably, reduced calories prevent brain atrophy and neurodegeneration, increase neurogenesis and improve mood disorders [37]. It is reported that calorie restriction changes the anxiety-like behavior of aging rats in a duration-dependent manner [38]. Moreover, Lu et al. also found that acute calorie restriction effectively defended against stress-related anxiety in mice [39]. In our study, we found that calorie restriction mitigated the obesity-associated anxiety in obese students. Moreover, we observed that more obese students exhibited an improvement of anxiety after calorie restriction as compared with the underweight students. Notably, the obese students who were on dieting were more likely to show an improvement of anxiety than the obese students who were not on the dieting (data not shown). Collectively, these findings indicate that calorie restriction could be an effective approach to improve anxiety, especially in obese college students. The information on anxiety management should be provided to college students, considering that the students with great anxiety may tend to overeat. Anxiety-associated overeating should be self-monitored and alternative anxiety-management strategies should be exercised for improving anxiety. On the other hand, the college administration and the University-affiliated student societies and clubs may help college students with self-monitor and anxiety management. The availability of a variety of healthy food options on campus including in cafes and vending machines, along with nutritional information, could protect the students from the imbalanced calorie intake. The balanced calorie intake may be an ideal and effective approach to reduce the anxious and obese levels of college students and should be followed in daily life.

There are some limitations in this study which can be improved in future research. As this study was conducted with the students attending a single college campus in China, a geographical limitation exists. Dietary habits and nutrient intake caused by different cultures would be expected to affect outcomes, which are not examined in this study. More college students should be included for further analysis. Additionally, whether or not the higher academic pressures may cause anxiety and anxiety-associated obesity needs further investigation. This study used the frequency of intake of particular foods to determine dietary behaviors, whereas neither the amount of consumed food nor caloric information was considered. In further research, standard scale will be utilized to detect dietary behaviors and calorie intake of college students. The assessments of sleep quality will also be assessed by Pittsburgh sleep quality index (PSQI). In this study, we found that the short-term calorie restriction (3–7d) can effectively improve the anxiety of college students. Further, a long-term intervention of calorie intake will be performed to examine its effects on anxiety among college students because the duration of intervention may affect the outcome of calorie intervention.

## 5. Conclusions

In this study, the college students displayed a positive association between anxiety and obesity. The unhealthy calorie intake was identified as a risk factor for anxiety, and the long sleep duration was identified as a protective factor for anxiety. It is important to investigate and understand the factors affecting anxiety, owing to the fact that anxiety can affect health and academic outcomes throughout adulthood of the college students. Support from college administrations such as healthy interventions is necessary to relieve the anxiety of the students. Supportive social relationships, positive family function and positive coping style may play an important role in reducing the stress and improving their mental well-being. By broadening social relationships and adopting more positive coping and less negative coping skills, anxiety symptoms may be prevented or at least diminished among college students.

Additionally, anxiety was relatively prevalent in the senior college students, suggesting that the higher academic pressure might be associated with the promotion of anxiety. Importantly, moderate calorie restriction can relieve anxiety in the obese students. Thus, the anxiety of college students would be managed by the promotion of healthy dietary behaviors to decrease excessive calorie intake. The administrations should take measures and interventions to reduce anxiety among college students and to provide educational counseling and psychological support for students to cope with these problems. Anxiety management needs to start before college life to avoid hindering the improvements of academic outcomes of college students.

## Figures and Tables

**Figure 1 nutrients-14-03518-f001:**
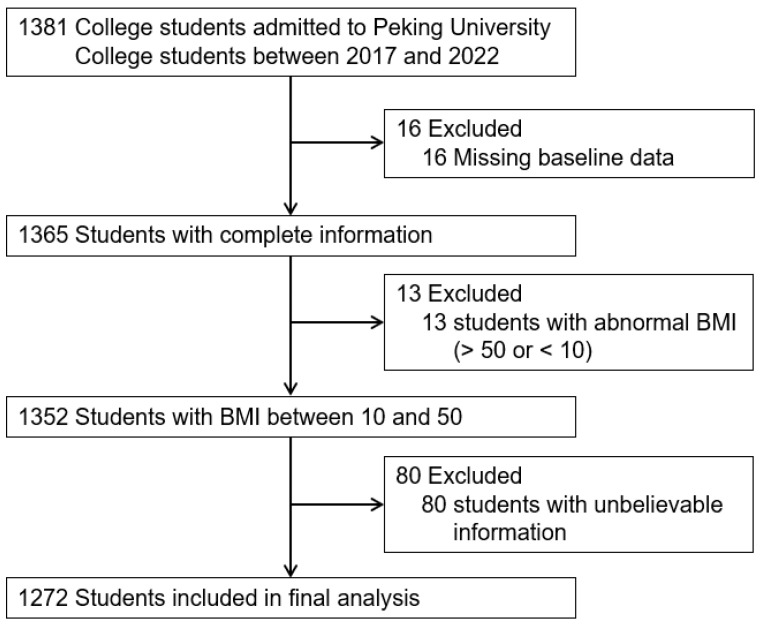
Flow diagram of inclusion and exclusion of study participants. BMI, body mass index.

**Figure 2 nutrients-14-03518-f002:**
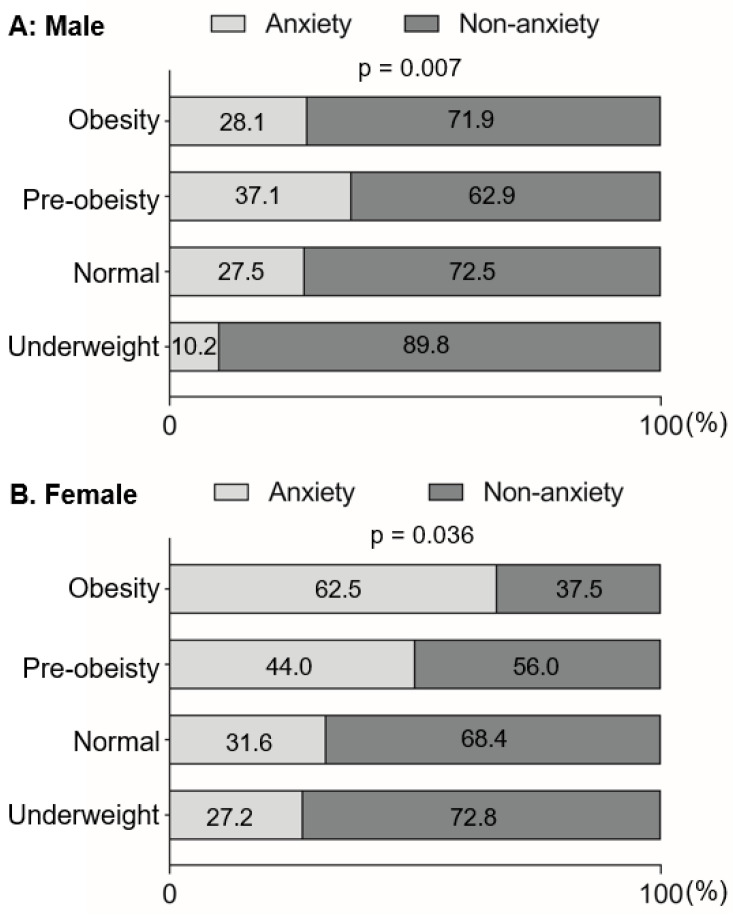
Baseline anxiety distribution among underweight, normal-weight, pre-obese and obese students in males (**A**,**B**) females. Dark and pale gray bars indicate the proportion of non-anxiety and anxiety, respectively.

**Table 1 nutrients-14-03518-t001:** Characteristics of respondents regarding demographics, dietary habits and anxiety (*n* = 1272).

Characteristics		Frequency	Percentage (%)
sex	Male	546	42.9
	Female	726	57.1
BMI	Underweight	211	16.6
	Normal weight	866	68.1
	Pre-obesity	155	12.2
	Obesity	40	3.1
Age	≤18	295	23.2
	19	404	31.7
	20	321	25.2
	21	174	13.7
	≥22	78	6.2
Academic year	Freshman	385	30.3
	Sophomore	531	41.7
	Junior	241	18.9
	Senior	115	9.0
Major	Non-medical science	762	59.9
	Medical science	510	40.1
BAI	Non-anxiety	889	69.9
	Anxiety	383	30.1
Sleep duration (h/d)	<4	31	2.4
	4–6	157	12.3
	6–8	900	70.8
	>8	184	14.5
Dieting	No	1104	86.8
	Yes	168	13.2
Calorie intake (kcal/d)	Healthy	576	45.3
	Unhealthy	696	54.7
Physical activity (MET∙min/w)	0–600	397	31.2
	600–3000	860	67.6
	>3000	15	1.2

**Table 2 nutrients-14-03518-t002:** Results of univariate logistic regression analysis for the association between anxiety and obesity.

Characteristics	*p* Valve	OR (95% CI)
Male		
Underweight	-	1
Normal weight	0.009	3.53 (1.36–9.12)
Pre-obesity	0.005	4.40 (1.57–12.33)
Obesity	0.040	5.03 (1.08–23.40)
Female		
Underweight	-	1
Normal weight	0.249	1.26 (0.85–1.86)
Pre-obesity	0.013	2.68 (1.24–5.82)
Obesity	0.175	5.36 (0.47–60.64)

**Table 3 nutrients-14-03518-t003:** Baseline characteristics of 1272 students stratified by anxiety and non-anxiety.

Anxious Status	All	Non-Anxiety	Anxiety	*p*
Number	1272	889	383	
Baseline characteristics				
Sex				0.126
Male	546	394	152	
Female	726	495	231	
BMI				0.009
Underweight	211	162	49	
Normal weight	866	607	259	
Pre-obesity	155	94	61	
Obesity	40	26	14	
Major				0.188
Non-medical science	762	522	240	
Medical science	510	367	143	
Calorie intake				<0.001
Healthy	576	432	144	
Unhealthy	696	457	239	
Physical activity				0.926
0–600	397	279	118	
600–3000	860	599	261	
>3000	15	11	4	
Sleep duration				<0.001
<4	31	13	18	
4–6	157	94	62	
6–8	900	643	257	
>8	184	138	46	
Academic year				0.096
Freshman	385	286	99	
Sophomore	531	366	165	
Junior	241	157	84	
Senior	115	80	35	
Dieting				0.002
No	1104	789	315	
Yes	168	100	68	

**Table 4 nutrients-14-03518-t004:** Results of univariate logistic regression analysis of anxiety.

Characteristics	*p* Valve	OR (95% CI)
Sex		
Male	-	1
Female	0.126	1.21 (0.95–1.54)
BMI		
Underweight	-	1
Normal weight	0.036	1.45 (1.02–2.06)
Pre-obesity	0.005	2.02 (1.23–3.31)
Obesity	0.107	2.48 (0.82–7.49)
Major		
Non-medical science	-	1
Medical science	0.19	0.85 (0.66–1.08)
Calorie intake		
Healthy	-	1
Unhealthy	<0.001	1.57 (1.23–2.01)
Physical activity		
0–600	-	1
600–3000	0.335	0.88 (0.68–1.14)
>3000	0.665	0.77 (0.24–2.48)
Sleep duration		
<4	-	1
4–6	0.06	0.47 (0.22–1.03)
6–8	0.001	0.29 (0.14–0.60)
>8	<0.001	0.24 (0.11–0.53)
Academic year		
Freshman	-	1
Sophomore	0.077	1.30 (0.97–1.75)
Junior	0.015	1.55 (1.09–2.19)
Senior	0.317	1.26 (0.80–2.00)
Dieting		
No	-	1
Yes	0.002	1.70 (1.22–2.38)

**Table 5 nutrients-14-03518-t005:** Results of multivariate logistic regression analysis of anxiety.

Variables	β	SE	*p* Valve	OR (95% CI)
BMI				
Underweight	-	-	-	1
Normal weight	0.336	0.182	0.065	1.40 (0.98–2.00)
Pre-obesity	0.648	0.257	0.012	1.91 (1.16–3.17)
Obesity	0.785	0.579	0.175	2.19 (0.71–6.82)
Calorie intake				
Healthy	-	-	-	1
Unhealthy	0.444	0.128	0.001	1.56 (1.21–2.00)
Sleep duration				
<4	-	-	-	1
4–6	−0.821	0.404	0.042	0.44 (0.20–0.97)
6–8	−1.337	0.376	<0.001	0.26 (0.13–0.55)
>8	−1.541	0.408	<0.001	0.21 (0.10–0.48)
Academic year				
Freshman	-	-	-	1
Sophomore	0.302	0.154	0.049	1.68 (1.19–2.36)
Junior	0.499	0.183	0.006	1.35 (1.00–1.83)
Senior	0.245	0.24	0.307	1.28 (0.80–2.04)
Dieting				0.002
No	-	-	-	1
Yes	0.517	0.175	0.003	1.68 (1.19–2.36)

**Table 6 nutrients-14-03518-t006:** Baseline characteristics of 230 students stratified by improvement and non-improvement of anxiety.

Improved Status	All	Non-Improvement	Improvement	*p*
Number	230	178	52	
Baseline characteristics				
BMI				0.001
Underweight	51	44 (86.3%)	7 (13.7%)	
Normal weight	159	123 (77.4%)	36 (35.9%)	
Pre-obesity	15	11 (73.3%)	4 (26.7%)	
Obesity	5	0 (0.0%)	5 (100%)	
Calorie intake				0.418
Healthy	82	61 (74.4%)	21 (25.6%)	
Unhealthy	148	117 (79.1%)	31 (20.9%)	
Academic year				0.727
Freshman	61	47 (77.0%)	14 (23.0%)	
Sophomore	112	84 (75.0)	28 (25.0%)	
Junior	38	31 (81.6%)	7 (18.4%)	
Senior	19	16 (84.2%)	3 (15.8%)	
Dieting				0.408
No	210	164 (78.1%)	46 (21.9%)	
Yes	20	14 (70.0%)	6 (30.0)	

## Data Availability

The data presented in this study are available on request from the corresponding author.

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
