# Peer review of "Obesity-Associated Anxiety Is Prevalent among College Students and Alleviated by Calorie Restriction"

_nutrients, 2022, doi:10.3390/nu14173518_

Round 1
Reviewer 1 Report
The authors have conducted scientific research about the usefulness of calorie restriction in obesity and obesity-associated in adolescents and highlighted the prevalence of this association.
Major,
Additional data is required:
-The abstract can be readjusted and shortened, and should include study type, the total number of participants included and per group, and statistically significant data (p-value).
-the time period of this study/ when these participants were selected and completed this questionnaire? (this year, last year..)
-After “60, 20, 40, 100, and 140” the scale of measurement should be included.
- Characteristics of respondents (Table 1) should include data about age (As education systems differ around the world, the university is also important as some programs take 3 to 5 years to complete, the students can be between 14-19, or 15–18 depending on the admission)
-Also, all the adolescents were healthy. (ex: young patients with diabetes are prone to obesity, anxiety, and/or depression)
-What is the author's definition of “calorie restriction” in this case, how many calories per day?
-From the 230 subjects elected for calorie restriction, how many were females and how many males?
-The authors mentioned in their abstract that “Obesity contributes to metabolic disorders and disturbs the neural functions”, I suggest including a small paragraph in your 4. Discussion about the molecular pathways behind “hypercaloric diets impair fear-regulatory mechanism” by which obesity can lead to anxiety.
-The authors mentioned also, that students were divided according to their major, the medical science group and the non-medical science group. There should be a brief discussion regarding this aspect that correlated with the results. Did the non-medical science group that had more subjects with anxiety (n=270 from Table 3.) hold any statistical significance, and is linked to the final results?
Minor,
In “2. Participants” after “students from Peking University” I suggest including the region/city/country.
From rows 131 to 147, these two paragraphs correspond more promptly to chapter 3. Measurements not in 4. Statistical analysis
Attention to writing there are many grammatical errors and missing punctuations, throughout the manuscript; Examples:
- This sentence needs rephrasing “Therefore, it is important for college 60 students to prevent obesity as well as its associated disorders in adolescents”.
-Row 64- correct to: “through a systematic review and meta-analysis study”
-Row 68 “that the rats underwent” missing pronoun
-Row 134 “That the total score is more than 45 indicates” needs correction.
-Row 145 “About major, the medical science includes”
-Row 263 “..students, it may be due to a..” missing prepositions
-Row 298 “Moreover, w observed that” needs correction and so on..
Check Journals guidelines about the correct form of introducing references.
Author Response
We thank all two reviewers for their constructive comments. We have revised the manuscript according to the review comments. Below, we provided a point-to-point response to major review critics. We hope the reviewers find the major concerns adequately addressed and the revised manuscript has been improved.
Point-to-point response
Reviewer #1
The authors have conducted scientific research about the usefulness of calorie restriction in obesity and obesity-associated in adolescents and highlighted the prevalence of this association.
Major,
Additional data is required:
-The abstract can be readjusted and shortened, and should include study type, the total number of participants included and per group, and statistically significant data (p-value).
Response: We thank the reviewer for this comment. As suggested by reviewer, we have re-adjusted and shortened the abstract. We have added the study type, the total number of participants and p-value in the revised abstract.
-the time period of this study/ when these participants were selected and completed this questionnaire? (this year, last year.)
Response: Thank you for your advice. This study was performed from January to March in 2021. We have added this information in the method section of the revised manuscript.
-After “60, 20, 40, 100, and 140” the scale of measurement should be included.
Response: Thank you very much for careful reading and thoughtful comments. Following your comments, we have included the scale of measurement, thank you. In this study, a unit of energy (kcal/d) was used; and the total score of calorie intake (kcal/d) was calculated as the total calorie intake of each student per day. We have added this information in the method section of the revised manuscript.
- Characteristics of respondents (Table 1) should include data about age (As education systems differ around the world, the university is also important as some programs take 3 to 5 years to complete, the students can be between 14-19, or 15–18 depending on the admission)
Response: Thank you very much for raising this critical point. Following your comments, we have accordingly added the age as one of the characteristics of respondents in Table 1 in the revised manuscript.
-Also, all the adolescents were healthy. (ex: young patients with diabetes are prone to obesity, anxiety, and/or depression)
Response: Thanks very much for your valuable suggestion. Indeed, the participants in this study are healthy. In principle, the participants with diabetes, bulimia, anorexia, thyroid dysfunctions, drug/substance abuse, smoking and alcoholism etc. were not included in the study. We have added the exclusion criteria in the method section of the revised manuscript.
-What is the author's definition of “calorie restriction” in this case, how many calories per day?
Response: We are thankful to the reviewer for providing such an important suggestion. In this study, students who received calorie restriction were prescribed a low-calorie diet (400 kcal lower than the caloric requirement per day). We have added this information in the results of revised manuscript.
-From the 230 subjects elected for calorie restriction, how many were females and how many males?
Response: Thank you very much for bringing up this point. Among the 230 subjects elected for calorie restriction, there were 122 females and 108 males. We have added this information in the results of revised manuscript. Thank you.
-The authors mentioned in their abstract that “Obesity contributes to metabolic disorders and disturbs the neural functions”, I suggest including a small paragraph in your 4. Discussion about the molecular pathways behind “hypercaloric diets impair fear-regulatory mechanism” by which obesity can lead to anxiety.
Response: We thank the reviewer for the guidance and appreciate the careful consideration of our study. Indeed, the calorie intake represents one of the major determinants of mental health and the vulnerability to injury and disease. Following your valuable comments, we have accordingly included a small paragraph in the Discussion of the revised manuscript.
Here, we also presented this small paragraph for you, please see it and the references in the below:
Evidence has showed that long-term high fat diet increases serum corticosterone levels and hippocampal expression of CYP11B1, which elevates HPA activity and then induces anxiety [1]. In addition, the gut microbiota influences obesity-related gut- and intestinal-based metabolic factors and affects anxiety-like behavior via the microbiota-gut-brain axis [2, 3], which could partly explain the co-morbidity of obesity and anxiety-like behavior. Notably, reduced calories prevent brain atrophy and neurodegeneration, increases neurogenesis and improves mood disorders [4].
- Dutheil, S.; Ota, K. T.; Wohleb, E. S.; Rasmussen, K.; Duman, R. S. High-Fat Diet Induced Anxiety and Anhedonia: Impact on Brain Homeostasis and Inflammation. Neuropsychopharmacology 2016, 41, 1874-1887.
- Cryan, J. F.; O'Riordan, K. J.; Cowan, C. S. M.; Sandhu, K. V.; Bastiaanssen, T. F. S.; Boehme, M.; Codagnone, M. G.; Cussotto, S.; Fulling, C.; Golubeva, A. V., et al. The Microbiota-Gut-Brain Axis. Physiol Rev 2019, 99, 1877-2013.
- Noble, E. E.; Hsu, T. M.; Kanoski, S. E. Gut to Brain Dysbiosis: Mechanisms Linking Western Diet Consumption, the Microbiome, and Cognitive Impairment. Front Behav Neurosci 2017, 11, 9.
- Mattson, M. P. Energy Intake and Exercise as Determinants of Brain Health and Vulnerability to Injury and Disease. Cell Metab 2012, 16, 706-722.
-The authors mentioned also, that students were divided according to their major, the medical science group and the non-medical science group. There should be a brief discussion regarding this aspect that correlated with the results. Did the non-medical science group that had more subjects with anxiety (n=270 from Table 3.) hold any statistical significance, and is linked to the final results?
Response: Thank you very much for bringing up this point. Indeed, it has been reported that the major of students may affect their anxiety levels. Honestly, we noticed that several studies showing that the non-medical students exhibit a significantly higher anxiety levels than the medical students [1, 2].
There may be more professional medical knowledge and better health awareness in the medical students compared to the non-medical students, and this may be beneficial for the prevention of anxiety. Nevertheless, in our research, the prevalence of anxiety among the non-medical students was 31.5% while the prevalence of anxiety was 28.0% among the medical students; and no statistical significance was observed.
We speculated that this discrepancy may be due to the strengthening of health education for the college students in China. We have added this paragraph in the discussion and also included the references in the revised manuscript.
- Bhat, U. S.; Amaresha, A. C.; Kodancha, P.; John, S.; Kumar, S.; Aiman, A.; Jain, P. A.; Cherian, A. V. Psychological Distress among College Students of Coastal District of Karnataka: A Community-Based Cross-Sectional Survey. Asian J Psychiatr 2018, 38, 20-24.
- Mirza, A. A.; Milaat, W. A.; Ramadan, I. K.; Baig, M.; Elmorsy, S. A.; Beyari, G. M.; Halawani, M. A.; Azab, R. A.; Zahrani, M. T.; Khayat, N. K. Depression, Anxiety and Stress among Medical and Non-Medical Students in Saudi Arabia: An Epidemiological Comparative Cross-Sectional Study. Neurosciences (Riyadh) 2021, 26, 141-151.
Minor,
In “2. Participants” after “students from Peking University” I suggest including the region/city/country.
Response: Thank you very much for your suggestion. Following your advice, we have added the information (region/city/country) in the method section of the revised manuscript.
From rows 131 to 147, these two paragraphs correspond more promptly to chapter 3. Measurements not in 4. Statistical analysis
Response: Thank you very much for bringing up this point. Following your comment, we have moved these two paragraphs to chapter 3 in the revised manuscript.
Attention to writing there are many grammatical errors and missing punctuations, throughout the manuscript; Examples:
- This sentence needs rephrasing “Therefore, it is important for college 60 students to prevent obesity as well as its associated disorders in adolescents”.
-Row 64- correct to: “through a systematic review and meta-analysis study”
-Row 68 “that the rats underwent” missing pronoun
-Row 134 “That the total score is more than 45 indicates” needs correction.
-Row 145 “About major, the medical science includes”
-Row 263 “students, it may be due to a..” missing prepositions
-Row 298 “Moreover, w observed that” needs correction and so on.
Response:: Thank you very much for careful reading and thoughtful comments. As suggested by reviewer, we have accordingly corrected these places in the manuscript. Thank you very much.
Check Journals guidelines about the correct form of introducing references.
Response: Thank very much for your careful reading and valuable suggestion. We have corrected the references according to the Journals guidelines.
Reviewer 2 Report
The problem, which is considered in the paper „Obesity-associated Anxiety Is Prevalent among the College Students and Alleviated by Calorie Restriction, is important in the research domain. The structure is preserved and extremely legible. The introduction contains a clearly described research problem and the literature review is systematized. However, the paper needs to be improved in several domains:
- In the abstract, also in the data section, the authors must mention the period in which the case study was conducted.
- The implications of this research should be emphasized.
- Please, compute a measure of internal consistency. I recommend you to use it Cronbach's alpha?
Also, I appreciate the work and effort put in.
Author Response
Response letter
We thank all two reviewers for their constructive comments. We have revised the manuscript according to the review comments. Below, we provided a point-to-point response to major review critics. We hope the reviewers find the major concerns adequately addressed and the revised manuscript has been improved.
Point-to-point response
Reviewer #2
The problem, which is considered in the paper „Obesity-associated Anxiety Is Prevalent among the College Students and Alleviated by Calorie Restriction, is important in the research domain. The structure is preserved and extremely legible. The introduction contains a clearly described research problem and the literature review is systematized. However, the paper needs to be improved in several domains:
-In the abstract, also in the data section, the authors must mention the period in which the case study was conducted.
Response: Thank you for your advice. This study was performed from January to March in 2021. We have added this information in the abstract and also the method section of the revised manuscript.
-The implications of this research should be emphasized.
Response: We are thankful to the reviewer for providing such a constructive suggestion. Indeed, we agree with you that the implications of this research should be emphasized. Following your suggestion, we highlighted these implications in the Conclusion section of the revised manuscript.
We also presented this Conclusion section here for you, please see it the below:
In this study, the college students displayed a positive association between the anxiety and the obesity. The unhealthy calorie intake was identified as a risk factor for anxiety, and the long sleep duration was identified as a protective factor for anxiety. It is important to investigate and understand the factors affecting anxiety, owing to the fact that anxiety can affect health and academic outcomes throughout adulthood of the college students. Support from college administrations such as healthy interventions is necessary to relieve the anxiety of the students. Supportive social relationships, positive family function and positive coping style may play important roles in reducing the stress and improving their mental well-being. By broadening social relationships and adopting more positive coping and less negative coping skills, anxiety symptoms may be prevented or at least diminished among medical students.
Additionally, the anxiety was relatively prevalent in the senior college students, suggesting that the higher academic pressure might be associated with the promotion of anxiety. Importantly, moderate calorie restriction can relieve anxiety in the obese students. Thus, the anxiety of college students would be managed by the promotion of healthy dietary behaviors to decrease excessive calorie intake. The administrations should take measures and interventions to reduce anxiety among college students and to provide educational counseling and psychological support for students to cope with these problems. Anxiety management needs to be started before college life to avoid hindering the improvements of academic outcomes of college students.
-Please, compute a measure of internal consistency. I recommend you to use it Cronbach's alpha?
Response: Thank you very much for bringing up this valuable point. Following your advice, we have calculated the internal consistency of the STAI. The Cronbach's alpha was 0.94; this indicates a high reliability of the responses. We have added this information in the results of revised manuscript. Thank you very much.
Round 2
Reviewer 1 Report
As seen, the authors have taken into consideration all reviewers observations, completed, and corrected the manuscript as suggested.